

# Trypanosoma cruzi reservoir—triatomine vector co-occurrence networks reveal meta-community effects by synanthropic mammals on geographic dispersal

Carlos N. Ibarra-Cerdeña[1], Leopoldo Valiente-Banuet[2], Víctor Sánchez-Cordero[3], Christopher R. Stephens[2,4] and Janine M. Ramsey[5]

[1] Departamento de Ecología Humana, Centro de Investigaciones y de Estudios de Avanzados (Cinvestav) del IPN Unidad Mérida, Mérida, Yucatán, México
[2] Centro de Ciencias de la Complejidad (C3), Universidad Nacional Autónoma de México, Ciudad de México, México
[3] Instituto de Biología, Universidad Nacional Autónoma de México, Ciudad de México, México
[4] Instituto de Ciencias Nucleares, Universidad Nacional Autónoma de México, Ciudad de México, Mexico
[5] Centro Regional de Investigaciones en Salud Pública, Instituto Nacional de Salud Pública, Tapachula, Chiapas, México

Corresponding authors
Carlos N. Ibarra-Cerdeña,
ibarra.cerdena@gmail.com,
cibarra@cinvestav.mx
Janine M. Ramsey, jramsey@insp.mx

## ABSTRACT

Contemporary patterns of land use and global climate change are modifying regional pools of parasite host species. The impact of host community changes on human disease risk, however, is difficult to assess due to a lack of information about zoonotic parasite host assemblages. We have used a recently developed method to infer parasite-host interactions for Chagas Disease (CD) from vector-host co-occurrence networks. Vector-host networks were constructed to analyze topological characteristics of the network and ecological traits of species' nodes, which could provide information regarding parasite regional dispersal in Mexico. Twenty-eight triatomine species (vectors) and 396 mammal species (potential hosts) were included using a data-mining approach to develop models to infer most-likely interactions. The final network contained 1,576 links which were analyzed to calculate centrality, connectivity, and modularity. The model predicted links of independently registered *Trypanosoma cruzi* hosts, which correlated with the degree of parasite-vector co-occurrence. Wiring patterns differed according to node location, while edge density was greater in Neotropical as compared to Nearctic regions. Vectors with greatest public health importance (*i.e., Triatoma dimidiata*, *T. barberi*, *T. pallidipennis*, *T. longipennis*, etc), did not have stronger links with particular host species, although they had a greater frequency of significant links. In contrast, hosts classified as important based on network properties were synanthropic mammals. The latter were the most common parasite hosts and are likely bridge species between these communities, thereby integrating meta-community scenarios beneficial for long-range parasite dispersal. This was particularly true for rodents, >50% of species are synanthropic and more than 20% have been identified as *T. cruzi* hosts. In addition to predicting potential host species using the co-occurrence networks, they reveal regions with greater expected parasite mobility. The Neotropical region, which includes the Mexican south and southeast, and the Transvolcanic belt, had greatest potential active *T. cruzi* dispersal, as well as greatest edge density. This information

could be directly applied for stratification of transmission risk and to design and analyze human-infected vector contact intervention efficacy.

# INTRODUCTION

Many of the most important and yet neglected human infectious diseases, such as Chagas Disease (CD), Leishmaniasis, Onchocerciasis, and Schistosomiasis (*Lozano et al., 2012*; *Murray et al., 2012*), have etiologic agents that are transmitted among multiple vector and reservoir species (*Taylor, Latham & Woolhouse, 2001*; *Heesterbeek et al., 2015*). Among these, the protozoan parasite *Trypanosoma cruzi* is responsible for human trypanosomiasis and Chagas Disease (CD). A century after its discovery (*Chagas, 1909*), CD continues to be among the main neglected tropical diseases in Latin America, based on mortality, disability-adjusted life years lost (DALYs), and at-risk population (*Hotez et al., 2008*). This chronic parasitic disease is the most frequent cause of heart failure in rural populations of endemic countries (*Rassi, Rassi & Marin-Neto, 2010*; *Bui, Horwich & Fonarow, 2011*), where primary parasite transmission occurs due to human contact with *T. cruzi* contaminated bug faeces, during the vector's bloodmeal.

The Kinetplastid, *Trypanosoma cruzi* (Trypanosomatidae) is an obligate parasite that alternates between invertebrates (kissing bugs belonging to the family Reduviidae) and terrestrial mammals. It has low species-host specificity along its geographic range in the American continent (*Izeta-Alberdi et al., 2016*), which may be explained by intrinsic factors related to sophisticated host defense response mechanisms (*Freire-de-Lima et al., 2012*; *Caballero et al., 2015*) and to extrinsic factors that ensure persistence and transmission in vector-host communities. These latter factors may be associated with vectors and mammal host contact rates which in turn depend on habitat disturbance (*Ramsey et al., 2012*; *Lopez-Cancino et al., 2015*; *Gürtler & Cardinal, 2015*). These host-compatibility and host-encounter filters define the potential spectrum of parasites in hosts (*Krasnov et al., 2008*). Whereas host-compatibility, with early evolution of the mechanism for *T. cruzi* host cell invasion (*Caballero et al., 2015*; *Jackson et al., 2016*), has deep phylogenetic roots, host-encounter filters are more labile and depend on community assemblages and host species' population dynamics at local levels (*Kribs-Zaleta, 2010*).

Contemporary patterns of land use and global climate change modify regional species pools in multiple ways: on the one hand species interactions are being lost via local extinctions, and on the other they are gained via invasions which have as yet unknown net effects on ecosystem functioning (*Dirzo et al., 2014*). The modification of species assemblages resulting from disrupted trophic networks can affect parasite dynamics via cascade effects such as increasing contact rates between competent reservoirs and vectors (*Young et al., 2014*). This is important to consider from a public health perspective, since species linked to zoonotic diseases may be positively affected by anthropogenic disturbance, such

as habitat fragmentation (*Rubio, Ávila-Flores & Suzán, 2014*), defaunation (*Young et al., 2015*), or land-use change (*McCauley et al., 2015*). An increase in host ranges due to altered suites of niche conditions may trigger new meta-community arrays (i.e., local communities linked by dispersal of a number of interacting species), thereby benefiting long-range parasite dispersal (*Suzán et al., 2015*). These changes will inevitably affect the geographic pattern of vector-reservoir interactions and generate the opportunity for parasite flux across landscapes (*González-Salazar & Stephens, 2012*).

Since ecological communities have complex patterns in space and time (*Bascompte, 2010*), studying species interactions requires an approach that includes their key properties. A relatively new paradigm to study community complexity comes from network theory (*Green & Sadedin, 2005*; *Bascompte, 2009*), in which communities are depicted as wiring graphs and species are nodes connected by links based on their biotic interactions (*Fortuna & Bascompte, 2008*). The network approach focuses on the pattern of node interactions, key components of the network architecture (*Bascompte & Stouffer, 2009*). This approach has enlightened epidemiology by identifying network properties such as node centrality (i.e., relative intensity of connections with other nodes in the network, interpreted as node importance), which is related to node heterogeneity and affects the probability of pathogen spread by a host (*Gómez, Nunn & Verdú, 2013*). Network heterogeneities related to modularity (i.e., the extent to which the networks are subdivided in different modules of interacting species) have been used to understand how muti-host systems affect parasite sharing (*Pilosof et al., 2015*). Network modularity is greater in more complex communities of host-parasite networks, with parasite dynamics related to host phylogeny. Tick-borne pathogens are influenced more by host habitat sharing, as compared to their phylogenetic proximity (*Estrada-Peña et al., 2015*), while host-parasite interactions are more associated with the latter. Hence, naturally disrupted pathogen transmission routes can be "connected" by synanthropic animals (i.e., those species that can exploit human-modified habitats; *McFarlane, Sleigh & Mcmichael, 2012*). Synanthropy is not necessarily a definitive state, since it may decline when few resources are available in anthropogenic habitats, and hosts may maintain sylvatic populations (*Shochat et al., 2006*). Irrespective of taxa, synanthropic species are most likely to be pathogen hosts (*McFarlane, Sleigh & Mcmichael, 2012*).

Most disease-risk networks are constructed from known pathogen-host contact interactions (*Craft & Caillaud, 2011*), while that for less documented or more complex networks may be constructed using species co-occurrence networks (*Stephens et al., 2009*), which are based on the concept that at coarse-grained scales, interspecific interactions are associated with species co-occurrences (*Gotelli, 2000*). Distributional patterns may reflect evolved ecological relationships that confer either symmetrical or asymmetrical benefits when species occupy the same environment at the same time (*Cazelles et al., 2016*), thereby affecting community structure and function (*DiMichele et al., 2004*). This is particularly true for specialist guild groups, such as blood-sucking insect disease vectors, and the parasites they disperse (*Lehane, 2005*). Whereas host community composition over minor spatial (several meters) and temporal (several years) scales typically fluctuates widely in response to local disturbance, observations over hundreds of meters to thousands of kilometers are more predictable and have persistent patterns (*Jackson & Erwin, 2006*). The

latter has been reported due to long-term stasis in species occurrences (*DiMichele et al., 2004*) primarily due to niche constraints (*Martínez-Meyer, Peterson & Hargrove, 2004*), or to the stability of biotic interactions (*Morris et al., 1995*).

*Trypanosoma cruzi* vectors are blood-sucking insects of the Triatominae (Reduviidae:Hemiptera) that feed on terrestrial vertebrates and are obligate sanguivores. While triatomines seem to be eclectic in their blood sources, they most frequently feed on terrestrial mammals (*Noireau, Diosque & Jansen, 2009*). A possible outcome of the triatomine-mammal interaction is the transmission of *T. cruzi*, a parasite unique to this class (*Kierszenbaum, Gottlieb & Budzko, 1981*). With the exception of a few reports of triatomine-host interactions in North America that list the mammal species from which bugs feed (*Ibarra-Cerdeña et al., 2009*), few empirical studies have provided evidence for triatomine host community assemblages (*Izeta-Alberdi et al., 2016*). The complexity of vector-reservoir assemblages associated with *T. cruzi* transmission (i.e., number and strength of species interactions) has been associated with a gamut of transmission conditions in natural sylvatic areas (*Jansen, Xavier & Roque, 2015*; *Lopez-Cancino et al., 2015*; *Izeta-Alberdi et al., 2016*), anthropogenic ecotones (*Ramsey et al., 2012*), and urban habitats (*Delgado et al., 2011*; *Ramsey et al., 2000*). Host use by triatomines is influenced by the habitat they colonize, and that host accessibility may be a major factor shaping the blood-foraging patterns of these bugs (*Rabinovich et al., 2011*). At a coarse-grain scale, their more common hosts could be species with the highest exposure probability, where exposure is defined as the level of geographic co-variation between every bug species and every potential host. If co-occurrence patterns between triatomines and mammals are indicative of a potential vector-host interaction, a significant interaction should occur for those species pairs that are confirmed blood sources of triatomines and/or mammal reservoirs of *T. cruzi*. We used this argument to model biotic interactions between *T. cruzi* vectors (triatomines) and their hosts (mammals). We analyze empirically identified or evidence-based interactions to evaluate a network model's performance and address the following questions: (1) is regional host species richness related to the frequency of vector-host geographic co-distributions? Specifically, what is the most likely host for a particular vector species? A resulting hypothesis for this latter question is that host species richness (i.e., blood source availability) drives the vector's geographic dependence. (2) are vector-host co-distributions related to the transmission of *T. cruzi*? Specifically, do vector-host co-distributions predict host-*T. cruzi* interactions? Our hypothesis is that known *T. cruzi* hosts will have stronger associations with vector species within their geographic range. (3) is there a relationship between vector-host geographic "interactions" and human exposure to *T. cruzi*? The hypothesis is that synanthropic hosts and vectors play, within their geographic range, a significant role in complex network topology.

## MATERIALS AND METHODS
### Vector and mammal reservoir data
Georeferenced data points for triatomines and terrestrial mammal collection sites in Mexico were used to model interactions. For triatomines, we compiled a database consisting of

all documented data collections since 1979 (publication of the first global monograph of Triatominae by Lent and Wygodzynsky which re-assigned many autochthonous Mexican species complexes), published in technical reports (Mexican National Health Secretary), scientific publications, or scientific collections (i.e., INSP, INDrE and IBUNAM), that includes 1600 data points for 26 Mexican triatomine species (*Ramsey et al., 2015*; doi: 10.5061/dryad.rq120). Mammal data points were retrieved from the Global Biodiversity Information Facility (GBIF; http://www.gbif.org), registered for Mexican collections (last accessed in February of 2011). We obtained 36,000 records for 396 mammal species, which were all used for network construction.

## Interaction model based on vector-mammal co-distributions

The methodology for the vector-host interaction inference we will use is that of (*Stephens et al., 2009*) which we describe here for completeness. Triatomine species hosts are a subset of all terrestrial mammal species present in their distribution range $I' \subseteq I$. Let the difference of the probabilities $P(B_i|\mathbf{I}'') - P(B_i)$ be the probability of the presence of a particular triatomine species ($B_i$) occurring given the presence of a particular mammal species ($\mathbf{I}'$). To calculate this probability, we used a grid size N of $0.25°$ for the cell size. The robustness of the results with respect to changes in the grid size were considered in *Stephens et al. (2009)*. The grid covers the area of all ecoregions where the species $B_i$ has been reported. We consider this probability relative to the null hypothesis that $B_i$ and $\mathbf{I}''$ are uncorrelated and hence $P(B_i|\mathbf{I}'') = P(B_i)$. Then, $P(B_i|\mathbf{I}'') - P(B_i) = (N_{Bi\ \text{and}\ \mathbf{I}'}/N_I') - (N_{Bi}/N)$, where $N_{Bi\ \text{and}\ \mathbf{I}'}$ is the number of spatial cells where there is a co-occurrence of the taxon $B_i$ and the taxon $\mathbf{I}'$ (the potential mammal host), and $N_I'$ is the number of cells where that host take their stated values (a Boolean presence/absence value), within the grid. $P(B_i|\mathbf{I}_k)$ is the number of co-occurrences of the taxa $B_i$ and $\mathbf{I}_k$, and allows us to define the most important statistical associations between these pairs of species distributions. However, since $P(B_i|\mathbf{I}_k)$ is a probability, it does not consider sample size. If $P(B_i|\mathbf{I}_k) = 1$, this may be the result of a coincidence of $B_i$ and $\mathbf{I}_k$ in one or 1,000 spatial cells, the latter being more significant statistically. To adjust for this difference, the following binomial test for statistical significance is used:

$$\varepsilon(B_i|I_k) = \frac{N_{I_j}(P(B_i|I_k) - P(B_i))}{(N_{I_j} P(B_i)(1 - P(B_i)))^{1/2}}. \tag{1}$$

Equation (1) measures the statistical dependence of $B_i$ on $\mathbf{I}_k$ relative to the null hypothesis that the distribution of $B_i$ is independent of $\mathbf{I}_k$ and randomly distributed over the grid. The sampling distribution of the null hypothesis is binomial, where every cell is given a probability $P(B_i)$ of having a point collection of $B_i$. The numerator of Eq. (1), is the difference between the observed number of co-occurrences of $B_i$ and $I_k$, and the expected number in the case where the distribution of point collections was obtained from a binomial with sampling probability $P(B_i)$. Since this is a stochastic sampling, the numerator must be measured in appropriate "units". As the underlying null hypothesis is that of a binomial distribution, it is natural to measure the numerator in standard deviations of this distribution, which is used as the denominator of Eq. (1). If $N_{B_j} \geq 5{-}10$, and $P(B_i)$ and

$P(B_i|\mathbf{I}_k)$ are not too close to zero or one, then a normal approximation for the binomial distribution is adequate, in which case $\varepsilon(B_i|I_k) = 1.96$ represents the standard 95% confidence interval. Naturally, as $\varepsilon$ values increase, the hypothesis of interaction becomes stronger. Note that such a statistical association does not necessarily prove that there is a direct "causal" interaction between the two taxa, rather it allows for a statistical inference that should be validated experimentally subsequently or assessed with independent studies from the literature.

If co-occurrence patterns between triatomines and mammals are indicative of a vector-host interaction, more significant $\varepsilon$ values should be found for those species' pairs that are confirmed blood sources of triatomines and/or mammal reservoirs of *T. cruzi*. As a model performance measure, we examine the $\varepsilon$ value for known interactions identified empirically and for those reported in the literature (see Table S1). Usually, these reports target specific triatomine species (i.e., the blood source of *Triatoma pallidipennis* as reported in *Mota et al. (2007)*, etc), therefore, model performance was evaluated using those species for which evidence is available. If the vector-host interaction is an important component of *T. cruzi* transmission, the $\varepsilon$ value should correlate with the probability for a mammal species to be a *T. cruzi* reservoir. To evaluate this, we listed all confirmed reservoir mammals and conducted the same analysis as described above.

### Description of regional availability of vector hosts and interaction patterns

Vector host availability was measured as the number of mammal species in the vector's distribution range (the grid). This number determines the potential for vector–host interactions and the opportunity for vector specialization. A simple statistical description of frequency distribution of host availability was conducted across vector species to analyze how vectors relate to mammal species richness. The same procedure was conducted to analyze the number of mammal species with which each vector had an epsilon value $\geq 1.96$. We extracted the mean and standard error from the group of mammals with which each vector had significant interactions to describe comparatively the strength of vector-mammal interactions.

### Validation of the interaction model

A literature search was conducted in Google Scholar for publications that report the presence of *T. cruzi* in Mexican wild mammals and in vector hosts using bloodmeal analysis. We used these records to evaluate the model performance by comparing the distribution frequency of historically confirmed mammal hosts grouped by epsilon range against the distribution frequency of all mammals with no reports of *T. cruzi* presence. This allowed us to describe the model's predictability across different epsilon values.

### Network construction

We used 26 Triatominae species and 396 mammal species (Table S1) for network vertices (nodes). Vertices were inter-connected representing significant triatomine-mammal geographic associations ($\varepsilon \geq 1.96$). All species were classified using binary variables associated

with their ecological tolerance for anthropogenic habitats (synanthropic vs. non-synanthropic) and with *T. cruzi* infection, the latter used for model validation (Table S1). To examine the geographic pattern of vector-mammal potential interactions, a network was projected onto a map of Mexico in which nodes were located according to their range centroid, which was determined as the geographic centroid calculated from the point collections of the node species. We used an automatic procedure to draw the network with a force-directed algorithm that assigns forces among the set of edges and nodes. Briefly, this algorithm emulates a system as if the edges were springs and the nodes were electrically charged particles. Forces are applied to the nodes, thereby pulling them closer together or pushing them further apart. This is repeated iteratively until the system comes to an equilibrium state i.e., their relative positions do not change from one iteration to the next, and the graph is then drawn. The equilibrium state represents all forces in mechanical equilibrium. In the final graph, the position of every node corresponds to their connectivity level, indicating that the most connected vertices are located in the center of the graph and vertices with fewer edges are in the periphery (*Kamada & Kawai, 1989*). In order to analyze geographic patterns of vector-mammal interactions relevant for Chagas disease risk, we used modular algorithms to identify network "community structure", which use centrality indices to identify community boundaries. Instead of constructing a measure for edges that are central to communities (as in hierarchical clustering), focus was placed on the least centralized edges that are "between" communities (*Newman, 2006*). Rather than constructing communities by adding the strongest edges to an empty vertex set, they were constructed by progressively removing edges from the original graph. Vertex betweenness was used as the centrality measure, since it reflects the node influence in the network. The betweenness centrality of a vertex *i* is defined as the number of shortest paths between pairs of other vertices that run through *i*. To find which edges in a network are between other pairs, we generalize betweenness centrality to edges and define the edge betweenness as the number of shortest paths between pairs of vertices. If there is more than one shortest path between a pair of vertices, each path is given equal weight so that the total weight of all of the paths is unity. If a network contains communities or groups that are only loosely connected by a few inter-group edges, then all shortest paths between different communities must attach to one of these edges. Thus, the edges connecting communities will have high edge betweenness. By removing these edges, we separate groups and reveal the underlying community structure of the graph (*Girvan & Newman, 2002*). We further analyzed the structure of the network by disentangling the relative effect of triatomines and mammals grouped according to their disturbance tolerance (synanthropic and non-synanthropic) and parasite association (*T. cruzi* and non-*T. cruzi*), in the network structure. To do this, we analyzed the mean degree of species belonging to each group. The degree of a node is the number of connections (edges) it has to other nodes in the network. Finally, we estimated the vector species' similarity in terms of their mammal host communities using a hierarchical cluster analysis based on the 1-Jaccard similarity coefficient (*Legendre, Borcard & Peres-Neto, 2005*). All analyses were conducted using the igraph package ver 1.0 for the statistical package R ver 2.15 (http://www.r-project.org). All routines were included in a script written to perform analyses (Supplemental Information 1).

## RESULTS

The matrix of 26 triatomine and 396 mammal species co-occurrence has 1,576 significant links from a total 2,695 (45%) possible links after taking into account that we only consider mammals present in the same projected ecoregions as a given vector (Table S2).

### Regional availability of vector hosts and interaction patterns

As would be expected, mammal species' number per vector was moderately correlated with vector grid size, i.e., the number of spatial grid cells occupied by a given vector (Spearman rank correlation coefficient $r = 0.61$, $P = 0.001$). The number of mammals to which vectors had significant epsilon values was less correlated with vector grid size ($r = 0.5$, $P = 0.01$). Six species of the *phyllosoma* complex (*Triatoma pallidipennis*, *Triatoma longipennis*, *Triatoma mazzottii*, *Triatoma picturata*, *Triatoma mexicana*, and *Triatoma gerstaeckeri*), one of the *protracta* (*Triatoma barberi*), and two of the *dimidiata* complex (*Triatoma dimidiata* Pacific hg2, and *T. dimidiata* Gulf hg3), had their distributional ranges which coincided significantly with regions having a high number of mammal species (i.e., higher than the 95% CI [79–128]). In contrast, eleven triatomine species (*Triatoma protracta*, *Triatoma peninsularis*, *Triatoma sinaloensis*, *Triatoma neotomae*, *Triatoma lecticularia*, *Triatoma nahuatlae*, *Triatoma uhleri*, *Triatoma zacatecensis*, *Triatoma woodi*, *Dipetatolagster maximus* and *Paratriatoma hirsuta*) had distribution ranges in regions with a significantly lower number of mammal species (i.e., lower than the 95% CI). These latter species, conversely, had a higher proportion of significant links (>95% CI) with mammal species with highest mean epsilon values (Figs. 1A and 1B). This pattern was inverted in vectors with high availability of mammals, since they had a lower proportion of accessible mammal hosts with significant links (Fig. 1B).

Rodentia and Chiroptera were the mammal orders with greatest number of species, however orders with the highest proportion of synanthropic species were Didelphimorphia and Rodentia, followed by Carnivora (Table 1). Didelphimorphia had the highest proportion of registered *T. cruzi* infected species, followed by Carnivora. Although Primates, followed by Rodentia, had greatest link strength (mean epsilon), rodents, opossums (Didelphimorphia), and bats (Chiroptera) had highest epsilon values. Opossums and Lagomorphs had the greatest number of links with vector species, although no Lagomorphs have been reported infected with the parasite.

### Vector-host co-occurrence and *Trypanosoma cruzi* transmission

All mammal species known to be *T. cruzi* hosts had significant epsilon values with at least one vector. The distribution frequency of host mammals was skewed positively with the magnitude of epsilon, in contrast to those mammals which have not been reported infected with *T. cruzi*. There was a significant difference between the proportion of *T. cruzi* hosts with high epsilon values, and the proportion of non-hosts with high epsilon (Fig. 2).

### Network topology and the importance of synanthropic hosts

The interactions wiring patterns differed depending on the spatial position of the geographic centroid of vector and host ranges (Fig. 3). Edge density is higher in the Neotropical region, in central and southeast Mexico, as compared to the Nearctic region in the
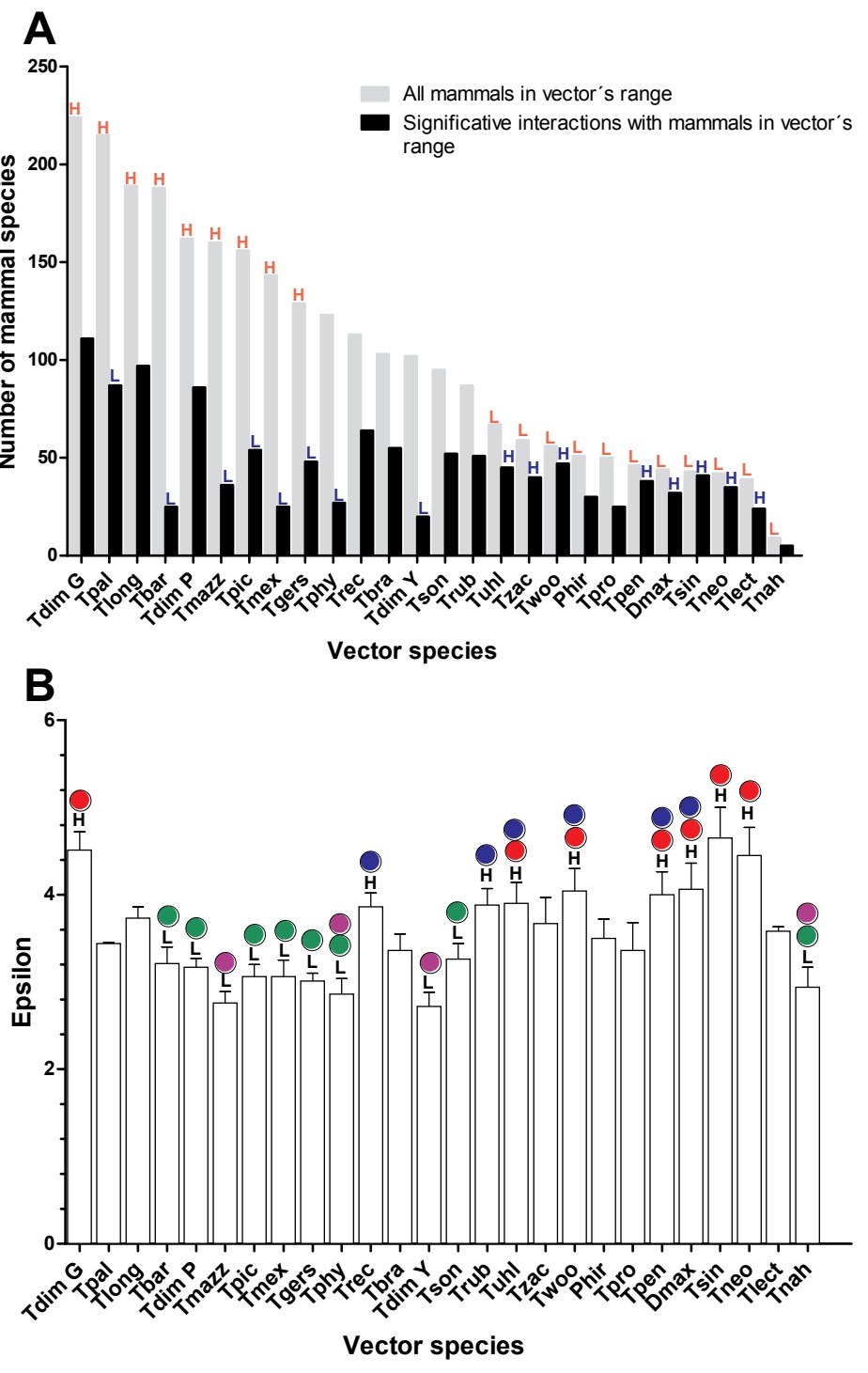

**Figure 1** **Relation between vector and mammal host species in their distributional range and their corresponding interaction strengths.** (A) Number of mammal species within each vector's range and the proportion of these with which the vectors had significant links ($\varepsilon \geq 1.96$). Red letters indicate bars that are above (H) or below (L), the 95% confidence interval (CI) of the average number of mammal richness per vectors' range. 

**Figure 1 (…continued)**
Blue letters indicate those bars that are above (H) or below (L) the 95% CI of the average proportion of mammal species with which vectors had significant links. (B) Mean and standard error of epsilon values for each vector species in their distribution ranges. Black letters indicate the species' vector that has mean epsilons outside of the 95% CI (H: higher and L: lower). Lack of statistical differences in the epsilon values are indicated for the same color with a $P < 0.05$. (Red and blue colors are used for lower epsilons, while green and purple are used for higher epsilons).

**Table 1** Summary of mammals atributes related with their relationships with vector co-occurrences and with Trypanosoma cruzi reports.

| Mammal order | Species | Synanthropic species | Reservoir species | Links per species | Epsilon | | |
|---|---|---|---|---|---|---|---|
| | N | N | N | | Mean | SE | Max |
| Artiodactyla | 7 | 0 | 0 | 2.6 | 3.4 | 0.29 | 6.06 |
| Carnivora | 29 | 11 | 9 | 4.9 | 3.95 | 0.15 | 13.65 |
| Chiroptera | 131 | 18 | 16 | 4.77 | 4.59 | 0.1 | 17.79 |
| Cingulata | 1 | 1 | 1 | 8 | 2.86 | 0.25 | 4.31 |
| Didelphimorphia | 6 | 4 | 4 | 5.3 | 4.91 | 0.63 | 19.42 |
| Lagomorpha | 8 | 0 | 0 | 5.5 | 4.84 | 0.4 | 13.48 |
| Pilosa | 2 | 0 | 0 | 4.5 | 4.02 | 0.84 | 10.25 |
| Primates | 3 | 0 | 0 | 4.5 | 6.4 | 1.48 | 10.83 |
| Rodentia | 170 | 94 | 34 | 3.82 | 5.78 | 0.13 | 24.84 |
| Soricomorpha | 15 | 0 | 0 | 2.73 | 4.91 | 0.44 | 12.96 |

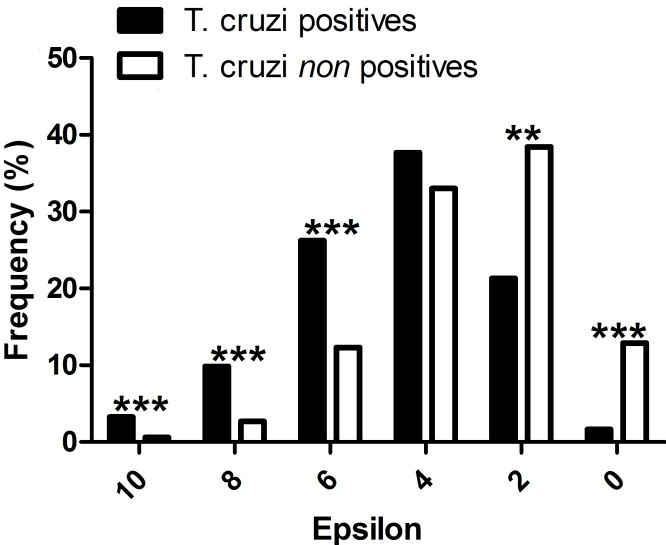

**Figure 2** Evaluation of the performance of the interaction model: *T. cruzi* potential host species are those mammals that were independently reported in the literature as testing positive for natural infections by *T. cruzi*. ***P-values < 0.001, and **P-values < 0.01.

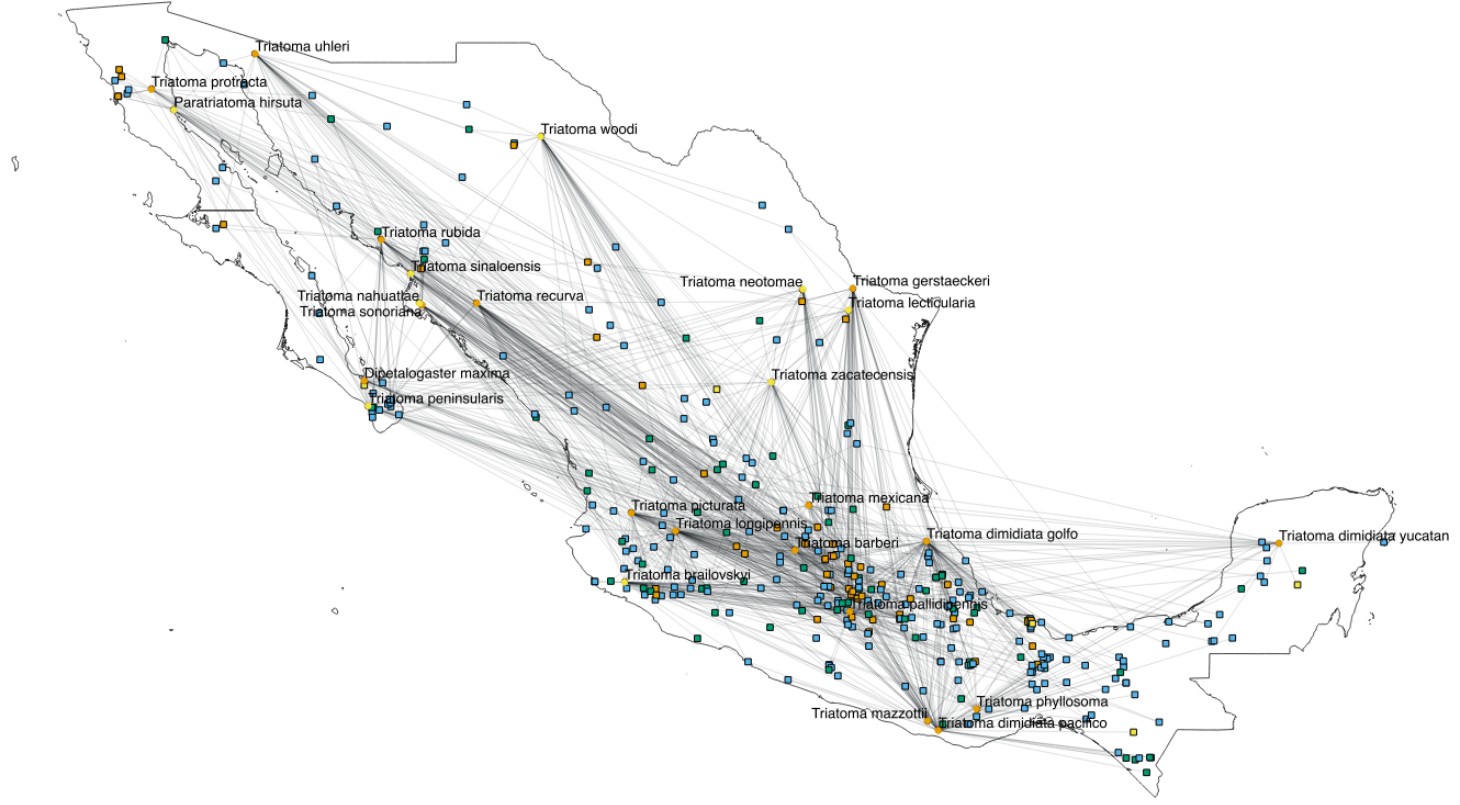

**Figure 3** **Network topology projected onto Mexico.** The network shows mammal (squares) and vector (circles) nodes linked through edges that represent significant geographic co-occurrence ($\varepsilon \geq 1.96$). Each mammal node is located in its range centroid within Mexico. Blue light are synanthropic mammals that are simultaneously infected and probable reservoirs of *T. cruzi*, yellow squares are synanthropic mammals that have not been reported infected with *T. cruzi*, orange squares are mammals that have been found infected with *T. cruzi* and are not synanthropic, and green squares are mammals that are neithers synanthropic nor have been reported with *T. cruzi* infection. Orange circles are synanthropic triatomines and yellow circles are non-synanthropic triatomines.

north. Synanthropic species (both triatomine and mammal) shape the Kamada–Kawai force-directed network structure, as indicated by their position in the center of graphs, while most non-synanthropic species are located on the periphery (Fig. 4). The structure of the complete array of mammals and vectors that can be considered relevant for the *T. cruzi* dispersal provided their geographic associations were not random, includes all significant links ($\varepsilon \geq 1.96$) (Fig. 4). Mammals which are both synanthropic and *T. cruzi*-infected are the most central in the network. These synanthropic mammals remained in greater proportion (compared to non-synanthropic mammals) in networks with higher epsilon values ($\varepsilon \gg 2$). In general, this last group of mammals (synanthropic and reported reservoirs), had proportionally more significant links than the group of registered reservoir species that are non synanthropic (OR = 1.17, 95% CI [1.01–1.4]). At highest epsilon values ($\varepsilon \geq 8$), the network connectivity was reduced to isolated clusters (Fig. 4), although these clusters had several nodes of synanthropic/*T. cruzi* reservoir species. For instance, *Triatoma dimidiata* Yucatan hg1 (node 365) had 21 links in the network with $\varepsilon \geq 10$, and 9 (43%) of those links were synanthropic and *T. cruzi* reservoirs.

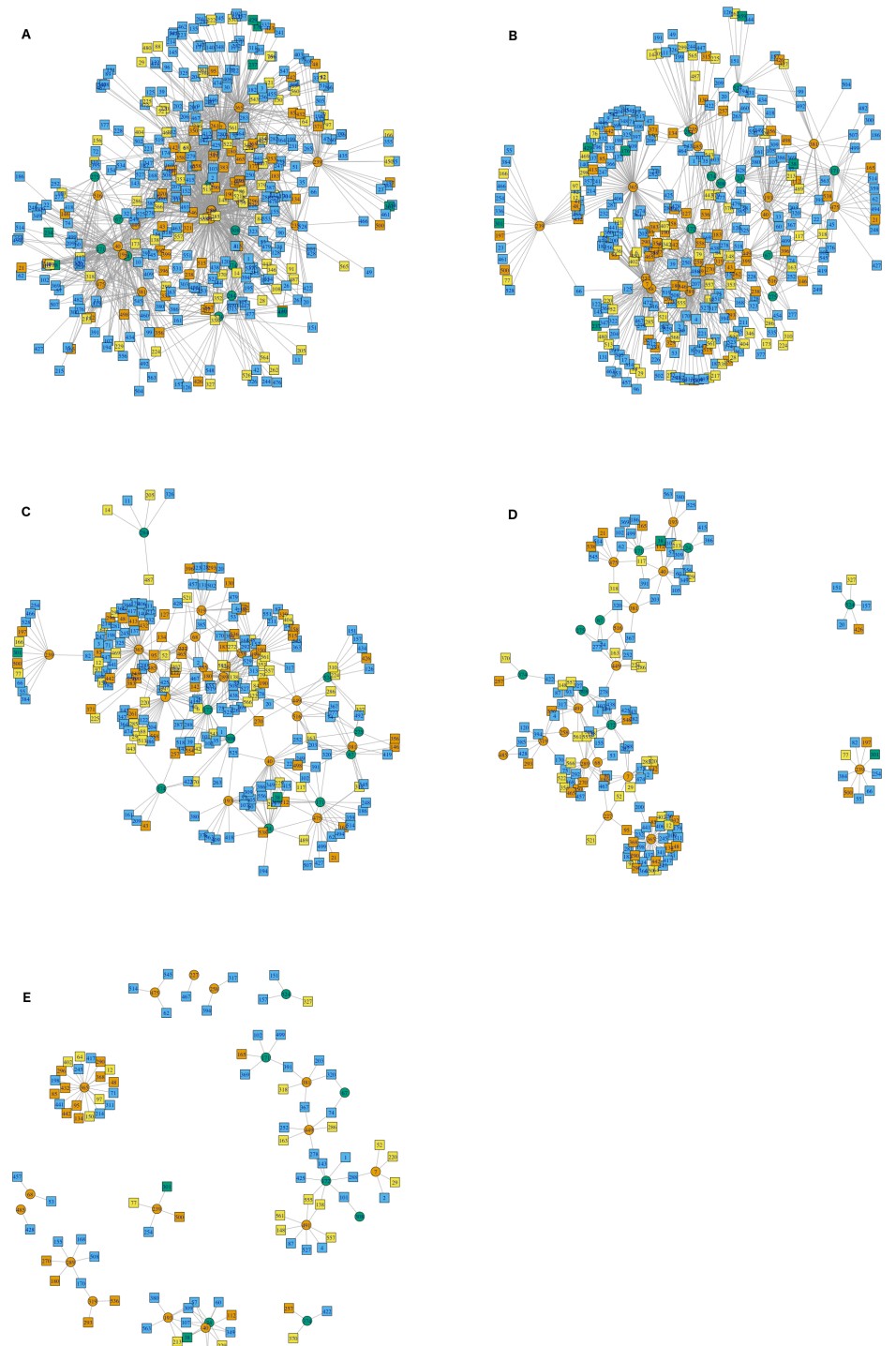

**Figure 4  Interaction network models for Mexican triatomines and terrestrial mammals.** Node position is determined with a Kamada–Kawai algorithm graph. Orange circles are synanthropic triatomines and green circles are non-synanthropic triatomines. Each network depicts resultant links when varying the epsilon threshold; (A) epsilon $\geq$ 1.96, (B) epsilon $\geq$ 4, (C) epsilon $\geq$ 6, (D) epsilon $\geq$ 8, and (E) epsilon $\geq$ 10. (continued on next page...)

**Figure 4 (…continued)**
Blue light are synanthropic mammals that are simultaneously infected and probable reservoirs of *T. cruzi*, yellow squares are synanthropic mammals that have not been reported infected with *T. cruzi*, orange squares are mammals that have been found infected with *T. cruzi* and are not synanthropic, and green squares are mammals that are neithers synanthropic nor have been reported with *T. cruzi* infection.

Network connectivity, as measured by node degree, decreased exponentially with an epsilon increase, both in vectors (Fig. 5A) and mammals (Fig. 5B). Synanthropic vectors were more connected than non-synanthropic ones (Fig. 5A). Similarly, synanthropic reservoirs of *T. cruzi* were more connected than other groups, particularly at low yet significant epsilon values (Fig. 5B). All groups had the same pattern of connectivity decrease with increasing epsilon, except for non-synanthropic reservoirs.

Two principal clusters emerged from the complete geographic model network. These clusters differed according to number of species and geographic region. The first cluster was composed principally of *phyllosoma* (seven species) and *dimidiata* (two haplogroups) complex species and 171 mammals, while the second cluster contained principally *protracta* and *rubida* complexes (17 vector species and subspecies) and had 183 mammals. The more dominant species of this latter group are *T. rubida*, *T. uhleri* and *T. woodii*. Several mammal and vector species were intermediate between these two former groups and could not be assigned to either: *T. dimidiata* hg1, *T. barberi*, *T. longipennis*, *T. mexicana,* and *T. pallidipennis* (Fig. 6). Hierarchical similarity of vectors constructed from mammal hosts produced two main clusters. Most synanthropic vectors were grouped into one cluster, while most non-synanthropic species were separated into a second cluster (Fig. 7).

## DISCUSSION

A spatial data-mining modeling approach, based on Complex Inference Networks (*Stephens et al., 2009*), has been used herein to predict vector–host interactions that could be meaningful for *T. cruzi* vector transmission and Chagas disease risk. Our results confirm that this approach can lead to accurate predictions of yet unidentified or unconfirmed species' interactions using public domain curated data, as has been demonstrated for predicting unknown hosts of *Leishmania mexicana* (*Stephens et al., 2009*; *Stephens et al., 2016*). This highlights the value of natural history museum and scientific collections to analyze ecological processes that affect human populations (*Sánchez-Cordero & Martínez-Meyer, 2000*). Our interaction model correctly predicted known mammal-parasite interactions using only geographic associations between infected vector and mammal reservoir species. Moreover, *T. cruzi* reservoirs most frequently occur within the highest rank of epsilon values. Since epsilon measures the statistical geographic association of vectors and mammals, the model additionally predicts interaction patterns which may influence parasite transmission and highlights how local interactions are affected by large-scale processes, previously demonstrated using spatial scaling models (*Brose et al., 2004*).

The geographic ecology of vector–host interactions in North American triatomines was first studied by (*Peterson et al., 2002*). These authors focused on the *protracta* complex, which had been reported to specialize on woodrats as a natural blood source (*Ryckman,*

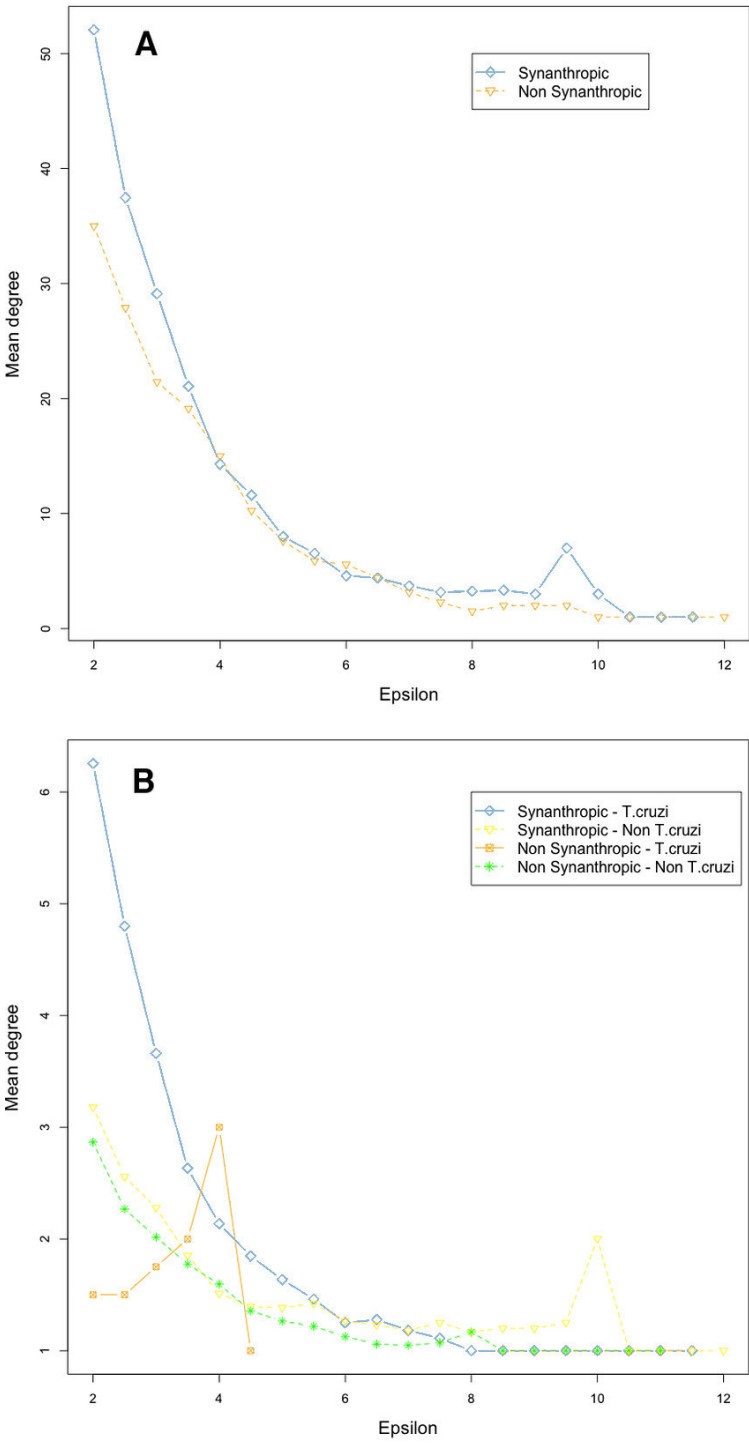

**Figure 5** Habitat affinities and *T. cruzi* relationships for vectors (A) and mammal hosts (B) in relation to node importance for the network structure.

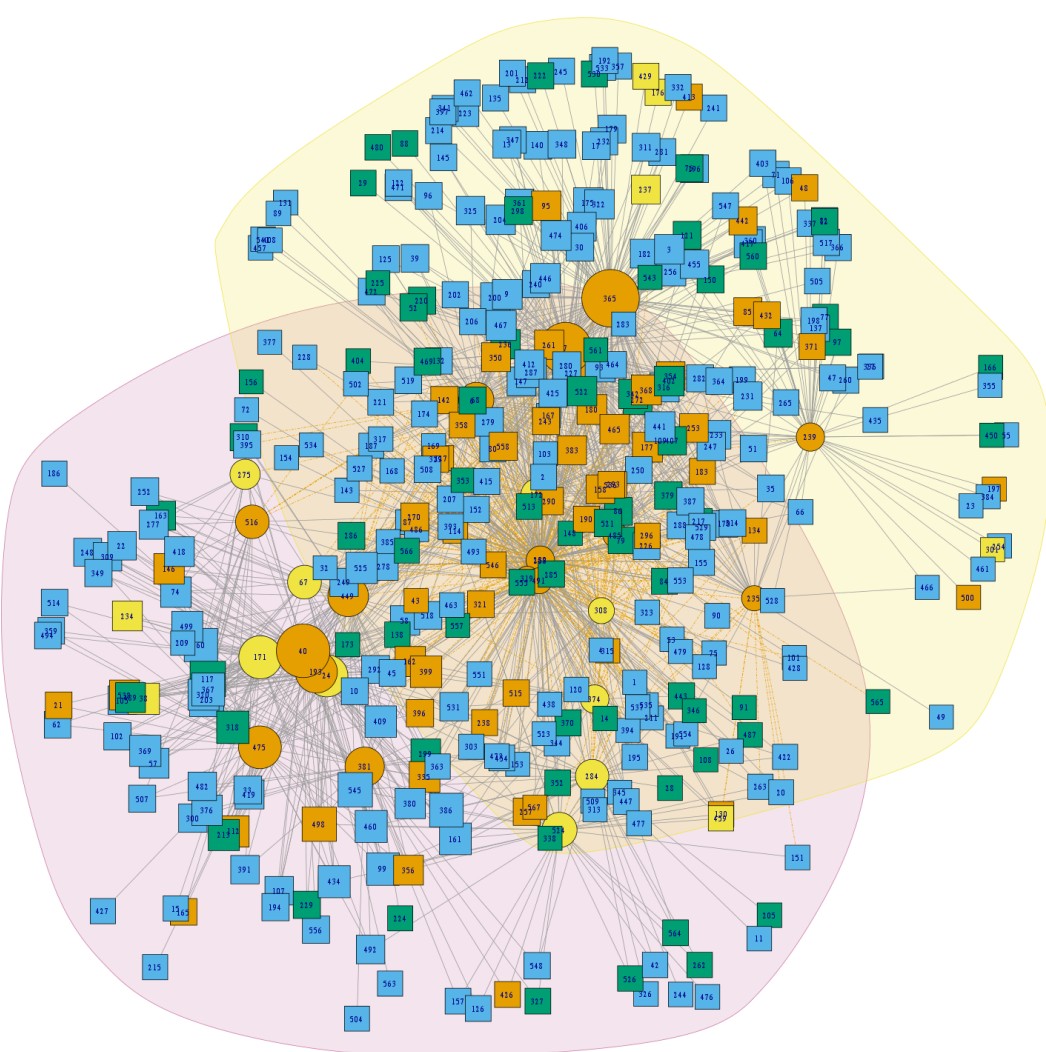

**Figure 6  Analysis of community structure in the triatomine-mammal host network model.** Node size indicates the proportional contribution of each species to its group. Circles represent vectors and squares mammal hosts. Polygones differentiate the two primary clusters, and yellow edges indicate the links that cause community interaction (bridge nodes). Orange circles are synanthropic triatomines and yellow circles are non-synanthropic triatomines. Blue light squares are synanthropic mammals that are simultaneously infected and probable reservoirs of *T. cruzi*, yellow squares are synanthropic mammals that have not been reported infected with *T. cruzi*, orange squares are mammals that have been found infected with *T. cruzi* and are not synanthropic, and green squares are mammals that are neithers synanthropic nor have been reported with *T. cruzi* infection.

*1986*). Using climatic niche modeling for hosts and vectors, associations for as yet unidentified hosts of *Triatoma barberi,* one of the primary *T. cruzi* vectors in Mexico, were predicted (*Peterson et al., 2002*). The present study extends the former to all Mexican vectors and to all potential terrestrial mammal *T. cruzi* reservoirs, by directly evaluating their potential biotic interactions. *Protracta* complex species occurring in both Nearctic and Neotropical regions, had significant epsilon values, with a high proportion of mammals inhabiting their distributional range sharing interactions with several species. This large-scale ecological

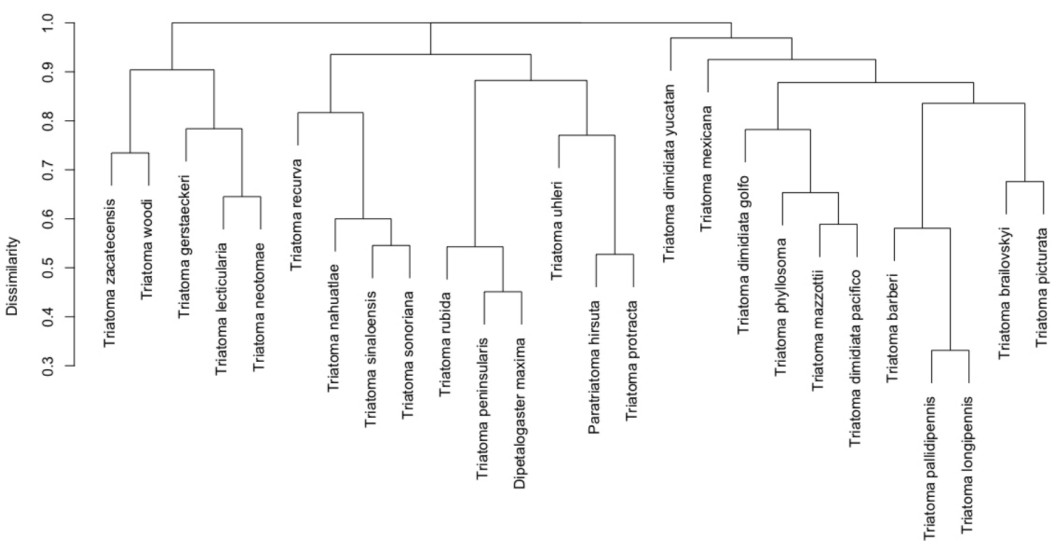

**Figure 7** Hierarchical clustering of mammal host similarity among vector species.

similarity could be causing a shift in host selection among *protracta* species, as confirmed by collection efforts (J Ramsey, pers. comm., 2016), an hypothesis addressed in lizard communities (*Losos et al., 2003*). In contrast, strictly Neotropical vector species, which are more permissive in host selection, have epsilon patterns consistent with low reliability on single hosts, and are associated with synanthropic mammals (*Rabinovich et al., 2011*; *Ramsey et al., 2012*).

Vectors occurring in intermediate network nodes (*T. dimidiata* hg1, *T. barberi*, *T. longipennis*, *T. mexicana,* and *T. pallidipennis*), between the two major communities, are responsible for more than 50% of *T. cruzi* transmission in Mexico (*Ramsey et al., 2015*). Hence, these species are important in generating a meta-community scenario within the network, and in maintaining parasite connectivity between the two other vector groups. Synanthropic mammals are keystone components of the geographic structure of triatomine-mammal network interactions. Although synanthropy is a relatively recent phenomenon, it signifies that fauna derive survival advantage from co-existence in human or human-related environments (*McFarlane, Sleigh & Mcmichael, 2012*). It is clear that the continuous provision of resources for mammal reservoirs, due to agriculture, food storage, garbage, livestock, land-use changes, and refuge, and hence triatomine blood resources year round, provide this fitness advantage (*McKinney, 2002*; *DeStefano & DeGraaf, 2003*; *Francis & Chadwick, 2012*). Expanding synanthropic populations are associated, therefore, with the degree of human landscape transformation (*Khlyap & Warshavsky, 2010*). The importance of synanthropic mammals as meta-community agents within landscapes in Mexico has been previously proposed (*Ramsey et al., 2012*; *Lopez-Cancino et al., 2015*). These authors suggested that disturbance-tolerant species, such as some rodent species, opossums, and bats, can connect host communities linked to specific habitats (i.e., conserved forests, cleared crop areas, or urban habitats). This provides an opportunity for *T. cruzi* flux within the landscape matrix over short periods, such as between seasons. Altered proportional

occupancy by parasite reservoirs due to habitat fragmentation, however, can provoke the loss of network connectivity (*Bordes et al., 2015*). If remaining species are highly mobile and tolerant to habitat disturbance, spatial connectivity of ecological interactions will increase. On a broader scale, major biogeographic regions in Mexico (Nearctic and Neotropical) can be connected by synanthropic species that persist after chronic disturbance, such as after land-cover changes for urbanization or agriculture. Although, the importance of such a mechanism over greater distance or longer periods has yet to be assessed, present results indicate that there is a biogeographic region association with triatomine-mammal interactions that depends on the specific community structure, connected by synanthropic species (*Izeta-Alberdi et al., 2016*).

Some species' groups associated with anthropozoonotic diseases have adapted to be almost completely synanthropic, such as dengue (*Lambrechts, Scott & Gubler, 2010*), or partially synanthropic, such as American trypanosomiasis (*Gaunt & Miles, 2000*). A gradient of synanthropy occurs in *T. cruzi* vector species in North America (*Ibarra-Cerdeña et al., 2009*), with highly synanthropic species, such as *T. pallidipennis* and most *phyllosoma* complex species (*Enger et al., 2004*; *Cohen et al., 2006*; *Ramsey et al., 2012*) and *T. dimidiata* haplogroups (*Dumonteil & Gourbiére, 2004*; *Guzmán-Tapia, Ramirez-Sierra & Dumonteil, 2007*; *Lopez-Cancino et al., 2015*), or partially synanthropic, such as *T. barberi* (*Ramsey et al., 2000*; *Martínez-Ibarra et al., 2008*). Reservoirs such as *Rattus rattus* or *Mus musculus* represent the case for highly synanthropic rodent species (*Khlyap & Warshavsky, 2010*), and *Sigmodon hispidus* or *Liomys irroratus* as partially synanthropic, all of them important reservoirs of *T. cruzi* (*Ramsey et al., 2012*; *Lopez-Cancino et al., 2015*). The present results are highly relevant for sustainable and cost-effective control of domestic or agricultural-related triatomine colonization, based on species' interaction scenarios with synanthropic reservoirs (*Peterson et al., 2002*; *Maher et al., 2010*). Evolutionary studies of *T. cruzi* diversification patterns indicate that long-term interaction with particular host species, may have had an influence on their phylogeny (*Yeo et al., 2005*). Most recent studies highlight the potential biogeographic impact of bats, their ability to disperse parasites over a greater range, and on their role in phylogenetic diversification of the *T. cruzi* clade (*Hamilton, Teixeira & Stevens, 2012*; *Lima et al., 2015*). We hypothesize that the large-scale network topology is critical for the geographic dispersal and diversification of *T. cruzi*, mediated principally by large-scale landscape transformation, and encourage research which traces *T. cruzi* population genetics and phylogeographic patterns using ecological network models as a template.

### Funding

Carlos N. Ibarra-Cerdeña CNIC was funded with a graduate scholarship from CONACYT (Consejo Nacional de Ciencia y Tecnología) for his PhD studies in the Biomedical Sciences Program of the UNAM (Universidad Nacional Autónoma de Mexico), fulfilled in part by this study. Janine Ramsey was funded by CONACyT FONSEC 161405 and CONACyT SEP

166828. Víctor Sánchez-Cordero was funded by PAPIIT-UNAM IN209314, Christopher R. Stephens was funded by PAPIIT-UNAM IN113414 y CONACyT Fronteras grant FC-2015-2/1093. The funders had no role in study design, data collection and analysis, decision to publish, or preparation of the manuscript.

### Grant Disclosures

The following grant information was disclosed by the authors:
CONACyT FONSEC: 161405.
CONACyT SEP: 166828.
CONACyT Fronteras: FC-2015-2/1093.
PAPIIT-UNAM: IN209314, IN113414.

### Competing Interests

The authors declare there are no competing interests.

### Author Contributions

- Carlos N. Ibarra-Cerdeña conceived and designed the experiments, performed the experiments, analyzed the data, contributed reagents/materials/analysis tools, wrote the paper, prepared figures and/or tables, reviewed drafts of the paper.
- Leopoldo Valiente-Banuet conceived and designed the experiments, performed the experiments, analyzed the data, contributed reagents/materials/analysis tools, prepared figures and/or tables, wrote R script.
- Víctor Sánchez-Cordero and Christopher R. Stephens conceived and designed the experiments, reviewed drafts of the paper.
- Janine M. Ramsey conceived and designed the experiments, contributed reagents/materials/analysis tools, wrote the paper, reviewed drafts of the paper.

### Data Availability

  The raw data has been supplied as a Supplementary File.

### Supplemental Information

Supplemental information for this article can be found online at http://dx.doi.org/10.7717/peerj.3152#supplemental-information.

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
