# Peer review of "Trypanosoma cruzi reservoir—triatomine vector co-occurrence networks reveal meta-community effects by synanthropic mammals on geographic dispersal"

_PeerJ, doi:10.7717/peerj.3152_

## Round 0.1 · original submission · Major Revisions

The manuscript consists of a considerable and interesting data set that adds an important contribution to the field. There are some points that need to be well addressed as detailed by the two expert reviewers. Please consider focusing the abstract highlighting the main results obtained. Please, also clarify the technical terms used and all the method strategies to make the paper clearer to the readers.

Reviewer 1 ·

Basic reporting

In general, the writing is very dense with ideas expressed in more complicated terms than necessary. Most of the ideas presented throughout the manuscript need to be unpacked and more clearly laid out, starting right from the title, which is very dense. Technical terms used throughout the manuscript should be defined.

Experimental design

No Comments

Validity of the findings

The results need to be given in more detail and much more thoroughly interpreted (and in plain English) rather than just stated. They are presented quite generally, and the reader is left to infer their meaning. For example- in the abstract the authors write that the 'network analyses allows us to predict not only potential host species but also regions with greater parasite mobility,' but then neither of these things are specifically reported in the paper- which mammal species? which bug species? which connections were most important? which geographic regions? These are the details that would make the paper interesting. Merely saying that synanthropic mammals have a major influence is not enough, and also, not breaking news.
Also, the figures are too busy and it is unclear what the authors intended to show with each figure, i.e., what the take home message is.

Additional comments

A stronger and more detailed justification for the work would improve the manuscript. How will this work and similar works fill holes/ help to solve challenges in Chagas disease ecology research? I'm sure there are benefits, but it was not clear.

Reviewer 2 ·

Basic reporting

This is a very good study on the use of network analysis in hosts-vectors interactions applied for Chagas disease. The analysis showed that synanthropic animals play and important role in the network architecture.
The methodology is good and based on a large database of mammals and the arthropod vectors of the trypanosome, agent of Chagas disease.
I have few comments, mainly concerning some missing definitions and potential references that the authors may consider.

In the Introduction:
the authors should give a definition of “synanthropy”, the authors can see this term intensively discussed in:
Khlyap LA, Warshavsky AA. 2010 Synanthropic and agrophilic rodents as invasive alien mammals. Russ J Biol Invasions 1:301–312.
McFarlane R et al. 2012. Synanthropy of wild mammals as a determinant of emerging infectious diseases in the Asian-Australasian region. EcoHealth 9, 24–35.

There are few references on the use of networks in epidemiology, I propose two among the many ones:
Gomez JM, Nunn CL, Verdu, M 2013 Centrality in primate-parasite networks reveals the potential for the transmission of emerging infectious diseases to humans. Proceedings of the National Academy of Sciences USA, 110, 7738-7741.
Pilosof S, Morand S, Krasnov BR, Nunn CL 2015 Potential parasite transmission in multi-host networks based on parasite sharing. PLoS ONE, 10, e0117909.

In Materials and Methods:
add the names of the packages used (and authors’ references)

in the discussion:
concerning the network architecture change in relation to habitat fragmentation:
Bordes F et al. 2015 Habitat fragmentation alters the properties of a host-parasite network: rodents and their helminths in South-East Asia. Journal of Animal Ecology 84, 1253–1263

supplementary data:
add the R scripts of the analyses

Experimental design

In Materials and Methods:
add the names of the packages used (and authors’ references)

supplementary data:
add the R scripts of the analyses

Validity of the findings

robust data, and results obtained with appropriate methologies

---

## Round 0.2 · accepted · Accept

All comments have been properly addressed.

Reviewer 2 ·

Basic reporting

I am very happy with the revised manuscript. The authors have done a good work.

Experimental design

excellent

Validity of the findings

good

Additional comments

he authors have done a good work in the revision of their manuscript. A paper that merits publication in Peer J